# The VP1u of Human Parvovirus B19: A Multifunctional Capsid Protein with Biotechnological Applications

**DOI:** 10.3390/v12121463

**Published:** 2020-12-18

**Authors:** Carlos Ros, Jan Bieri, Remo Leisi

**Affiliations:** Department of Chemistry and Biochemistry, University of Bern, 3012 Bern, Switzerland; jan.bieri@dcb.unibe.ch (J.B.); remo.leisi@dcb.unibe.ch (R.L.)

**Keywords:** parvovirus B19, B19V, VP1u, receptor, PLA_2_, virus entry, erythroid cells, biomarker, drug delivery, nanocarrier

## Abstract

The viral protein 1 unique region (VP1u) of human parvovirus B19 (B19V) is a multifunctional capsid protein with essential roles in virus tropism, uptake, and subcellular trafficking. These functions reside on hidden protein domains, which become accessible upon interaction with cell membrane receptors. A receptor-binding domain (RBD) in VP1u is responsible for the specific targeting and uptake of the virus exclusively into cells of the erythroid lineage in the bone marrow. A phospholipase A_2_ domain promotes the endosomal escape of the incoming virus. The VP1u is also the immunodominant region of the capsid as it is the target of neutralizing antibodies. For all these reasons, the VP1u has raised great interest in antiviral research and vaccinology. Besides the essential functions in B19V infection, the remarkable erythroid specificity of the VP1u makes it a unique erythroid cell surface biomarker. Moreover, the demonstrated capacity of the VP1u to deliver diverse cargo specifically to cells around the proerythroblast differentiation stage, including erythroleukemic cells, offers novel therapeutic opportunities for erythroid-specific drug delivery. In this review, we focus on the multifunctional role of the VP1u in B19V infection and explore its potential in diagnostics and erythroid-specific therapeutics.

## 1. Introduction

The *Parvoviridae* is a family of nonenveloped viruses that packages a linear, single-stranded DNA genome (~5 kb) within a small (~25 nm) icosahedral capsid. As a direct consequence of their limited coding potential, parvoviruses are particularly dependent on host cellular factors for their replication [1,2]. Parvoviruses are widely spread in nature and their host range might span the entire animal kingdom [3]. Depending on their host, members of the family *Parvoviridae* are subdivided into the subfamilies *Parvovirinae*, infecting vertebrates and *Densovirinae*, infecting insects and other arthropods. Viruses that infect vertebrates, including humans, are further divided into the dependoparvoviruses and the autonomous parvoviruses [4]. The dependoparvoviruses replicate only in the presence of a helper virus, such as adenovirus or herpesvirus. The adeno-associated viruses (AAVs) are not linked with any known pathology, have a wide tissue specificity, and replicate in dividing and nondividing cells. These properties make AAVs useful gene transfer vehicles for therapeutic applications [5]. Although autonomous parvoviruses use similar strategies for cell entry and replication, they differ substantially in their pathogenic potential, which ranges from subclinical to severe or even lethal infections [2]. As autonomous parvoviruses can only replicate in dividing cells, when the host cell DNA replication machinery becomes available, they tend to cause more severe infections in young than in adult hosts.

While most ssDNA viruses show a circular genome structure, parvoviruses have a linear genome, that is typically organized in two open reading frames (ORFs). The ORFs are flanked by palindromic sequences of variable length, which fold into hairpin structures and are essential for replication [6,7]. The 5′ ORF (ns or rep gene) encodes for the regulatory nonstructural protein(s) required for viral DNA replication and packaging. The 3′ ORF (cap gene) encodes two to four variants of a single capsid protein (VP). Following a principle of genetic economy, the different VPs are generated by alternative splicing or alternative codon usage, but also by post-translational proteolytic processing during entry, resulting in a common C-terminal sequence but different N-terminal extensions of variable length [8,9,10]. The T = 1 icosahedral parvovirus capsid is assembled from 60 VPs, however, the number of N-terminal VP variants used to assemble the infectious particles varies from two (VP1 and VP2) to four (VP1–VP4) depending on the genus. The VP variants are numbered in order of length, with VP1 being the largest variant. The common C-terminal region of the VPs forms the capsid shell, which consists of a conserved alpha-helix and a jelly roll motif containing eight antiparallel β-strands. The different configurations of the loops connecting the conserved β-strands delineate the surface topology, which is characteristic to each parvovirus genus and define the virus tropism and antigenicity [11]. Despite low sequence identity, the parvovirus capsids display structural features that are conserved across different genera, i.e., a narrow depression at the twofold axis of symmetry, protrusions of variable size and shape at the threefold axis and a canyon-like structure encircling a cylindrical pore at the fivefold axis connecting with the interior of the capsid.

The minor protein VP1 has an N-terminal extension of variable length, the so-called VP1-unique region (VP1u), and is present at 3 to 10 copies per virion depending on the parvovirus genus. VP1u is not required for virus assembly but contains several essential motifs required for the infection. Nuclear localization signals (NLSs) consisting of a stretch of basic amino acids have been identified in the VP1u from several parvoviruses. These motifs were shown to confer nuclear import potential to the incoming particles [12,13,14,15,16,17,18,19]. Another motif found in VP1u, except for amdoparvoviruses, is a phospholipase A_2_ (PLA_2_) enzyme domain, which enables viruses to escape from endosomal vesicles into the cytosol during cell entry [20,21,22,23,24,25,26]. Other motifs in VP1u were found to be essential for the infection. In AAV these motifs include signals that are known to be involved in protein interaction, endosomal sorting, and signal transduction in eukaryotic cells [27]. In B19V, a receptor-binding domain (RBD) required for virus uptake was identified at the N-terminal of the VP1u [28].

To infect the cell, parvoviruses follow an intricate path from the cell surface to the nucleus where they deliver the viral DNA for replication. During the process of entry, the incoming parvovirus capsids undergo a program of conformational rearrangements triggered by specific cellular factors that facilitate their intracellular transport [29,30]. A major capsid rearrangement that is largely conserved among parvoviruses involves the externalization of the VP1u region. Initially sequestered in mature virions, VP1u and its essential motifs become accessible at the particle surface during entry triggered by the acidic endosomal environment [10,16,31]. Besides low pH, AAVs may require additional cellular factors [31]. An exception is B19V, whose VP1u becomes accessible during the initial interactions with cellular receptors [32]. VP1u exposure occurs through the five-fold channel that connects with the interior of the capsid [33,34,35,36]. Structural and in vitro studies suggest that these channels serve not only as portals for the externalization of N-terminal capsid protein sequences but also for the packaging and release of the viral genome [16,31,37,38,39,40,41]. Mutations that perturb the functional structure of the channel result in defective genome encapsidation, uncoating and VP1u externalization [38,42,43]. B19V represents again an exception as the VP1u seems to be already exposed on the capsid surface, although in a conformation that is not accessible to antibodies [44].

This review focuses on the VP1u of B19V, which shares common aspects with other parvoviruses but has unique features, like its structural conformation relative to the virion, immunodominance, extraordinary length or the presence of a receptor-binding domain responsible for the restricted tropism of the virus. We present the current knowledge on the different VP1u motifs, their functions in the virus infection and the potential biotechnological applications of the B19V VP1u in human therapy and diagnostics.

## 2. Human Erythroparvovirus B19 (B19V)

B19V is the most prominent and well-characterized human pathogen within the *Parvoviridae* causing a mild childhood rash disease named *erythema infectiosum* or fifth disease [45]. The infection is often asymptomatic; however, in adults, B19V infection may induce a wide range of more severe pathological conditions, such as arthralgias and arthritis [46]. B19V infection may lead to aplastic crisis in patients with pre-existing bone marrow disorders and shortened red cell survival [47] and persistent infection in immunocompromised persons. Infection during pregnancy may result in *hydrops fetalis* and fetal death [48]. B19V was the first parvovirus known to cause disease in humans [49]. Since 2005, other human parvoviruses have been identified and include human bocavirus (HBoV1-4), parvovirus 4, bufavirus, tusavirus and cutavirus. Except for HBoV, which has been implicated in acute respiratory tract infections [50], the rest are emergent human parvoviruses with uncertain clinical significance [45,51].

B19V is transmitted via aerosol droplets that come into contact with the upper respiratory tract mucosa [47]. The virus crosses the mucosal epithelium through a yet unknown mechanism and disseminates with the bloodstream to the bone marrow, where it infects erythroid precursors at a particular erythropoietin (EPO)-dependent stage of differentiation [52,53,54]. The extraordinary narrow tropism of B19V is mediated at different levels of the viral life cycle. Crucial steps of the viral infection, such as uptake, genome replication, transcription, splicing and packaging, are restricted to the EPO-dependent erythroid differentiation around the proerythroblast stage [54,55,56,57,58,59,60]. The lytic replication cycle results in the destruction of the erythroid precursor cells [61,62], which accounts for the hematological syndromes observed during the infection [47]. Acute infection frequently results in high-titer viremia, which precedes the onset of clinical manifestations and has been associated with B19V transmission through transfusion and plasma-derived medicinal products [63].

## 3. B19V Capsid

The ssDNA genome of B19V is packaged into a small, nonenveloped, T = 1 icosahedral capsid. Similar to the genome of dependoparvoviruses, the B19V genome has two identical inverted terminal repeats (ITRs; ~383 nt), which serve as the origin of replication [64]. The capsid consists of 60 structural subunits of two N-terminal VP variants, VP1 and VP2. Approximately 95% are VP2 (major VP; 60 kDa) and 5% are VP1 (minor VP; 86 kDa) [65]. VP1 and VP2 are generated through alternative splicing, resulting in the same C-terminal sequence but VP1 contains 227 additional residues at the VP1 N-terminal region, the so-called VP1 “unique region” (VP1u). The 60 protomers form 20 trimeric capsomers in the cytoplasm of the infected cell, which are assembled to an icosahedral capsid structure in the host nucleus. Due to the T = 1 symmetry, all protein subunits can be assembled in the same orientation to each other. This perfect symmetry enables an optimal thermodynamic sink for each protomer interaction, forming a very stable capsid around the ssDNA genome.

Large-scale propagation of native B19V is not possible due to the lack of a fully permissive cell culture system. Accordingly, structural studies have been performed with recombinant B19V-like particles, which are similar, although not identical, to infectious native capsids. The structure of the VP2 recombinant particle has been determined to ~3.5 Å resolution [36]. Similar to other parvoviruses and many icosahedral viruses, the major capsid protein VP2 is structured as a “jelly roll” with a β-barrel motif. The loops connecting the strands of the β-barrel define the capsid surface topology that differentiates B19V from other parvoviruses. B19V lacks the prominent protrusions at the icosahedral threefold axes characteristic in other parvoviruses. The channel at the fivefold icosahedral axis is surrounded by a large canyon-like depression. Different from other parvoviruses, the channel in B19 VP2 capsids is constricted at its outside end. However, a cluster of glycine residues at this position may confer sufficient flexibility to open the channel upon specific cellular triggers during the infection. A striking difference between B19V and other parvoviruses is the external position of the N-VP2 and probably also VP1u [36,66]. Accordingly, the role of the fivefold channel in B19V would be limited to the externalization and packaging of the viral genome.

## 4. VP1u Is the Immunodominant Region of the Capsid but It Is Not Accessible in Native Virions

Although VP1u may occupy a surface position in the B19V capsid, different regions of the protein were shown to be inaccessible to antibodies. However, exposure of native capsids to heat or low pH rendered these regions accessible without capsid disassembly. In contrast to native virions, VP1u is always accessible in recombinant B19V-like particles [44]. The inaccessibility of VP1u in native virions is not well understood and may be explained by a compact structural conformation, or by the presence of a masking structure hiding the essential protein domains to the immune system. Despite the non-accessible conformation of VP1u in native particles, this protein is the immunodominant part of the capsid and contains clusters of critical neutralizing epitopes [67,68] (Figure 1).

Typically, neutralizing antibodies prevent the viral infection by interfering with early steps of the viral life cycle, i.e., attachment to cellular receptors, uptake, fusion, or conformational changes required for entry [69,70]. Importantly, the inhibition by neutralizing antibodies should be distinguished from the opsonization of viruses by antibodies, which can hamper the viral infection by immobilization of the virus and subsequent degradation of the immune complex by the complement system, immune cells or also by the cytoplasmic TRIM21/proteasome mechanism [71]. However, the specific targeting of essential capsid protein domains by neutralizing antibodies is required to efficiently interfere with the viral infection. Upon B19 viremia, the humoral immune response first generates IgM antibodies, which predominantly target the major capsid protein VP2. With the class-switch and long-term immunity, an increasing percentage of B lymphocytes secrete neutralizing antibodies against the VP1u region [72]. In this regard, a deficient immune response to VP1u has been associated with persistent infections, emphasizing the important role of the immune response against VP1u in clearing the virus [73,74].

Immunization experiments with vaccine candidates based on virus-like particles (VLPs) demonstrated that VP1u is essential to raise a strong neutralizing response against B19V [75,76]. However, the neutralization mechanism of antibodies targeting VP1u has remained largely elusive. Being the immunodominant region of the capsid, the originally inaccessible VP1u should become exposed in the extracellular milieu, and not inside endosomes, as shown for other parvoviruses. In line with this assumption, it has been shown that a neutralizing antibody against the N-terminal part of the VP1u was unable to bind native cell-free virions but was able to block virus entry into susceptible cells. Moreover, capsids without VP1u were unable to internalize into susceptible cells, demonstrating the involvement of the VP1u in B19V uptake [32,54]. These findings explain the high neutralization potential of VP1u antibodies, which target exclusively capsids during the initial interaction with cell receptors and block virus uptake, and further emphasize the importance of VP1u as an essential component of prospective B19V vaccines.

## 5. Role of VP1u in the Restricted Tropism of B19V

B19V has a remarkable narrow tropism. The virus shows productive infection exclusively in erythroid precursor cells at EPO-dependent intermediate erythroid differentiation stages, with increasing permissiveness from BFU-E to erythroblasts [53]. Viral tropism can be determined already at the cell surface by the expression of specific cell receptors required for virus entry and/or intracellularly by receptor-independent post-entry replication steps. The marked erythroid tropism of B19V is determined at multiple steps, i.e., the receptor-mediated uptake, genome replication, transcription, splicing and packaging [56,57,77]. A virus requiring such strict intracellular conditions for replication would also require a selective mechanism of cell entry to target exclusively the few cells where the virus can replicate. This strategy would allow the virus to avoid internalizing non-permissive cells, which would lead to abortive infections and inefficient viral propagation. Accordingly, it would be expected that B19V uses an erythroid-specific surface molecule as an entry receptor.

### 5.1. VP1u Contains a Receptor-Binding Domain That Is Essential for Virus Entry into Permissive Cells

The neutral glycosphingolipid globoside (Gb4), also known as P antigen, has long been considered the primary receptor of B19V [78]. A large body of evidence suggests that B19V recognizes Gb4 and that the interaction is required for the infection [79,80,81,82]. However, the wide-range Gb4 expression does not correlate well with B19V binding and uptake and cannot either explain the pathogenesis or the remarkable narrow tissue tropism of the virus [83]. By using a knockout cell line, we demonstrated that Gb4 does not have the expected function as the primary cell surface receptor required for B19V entry. Instead, Gb4 has an essential role at a post-entry step after virus uptake and before the delivery of the viral genome into the nucleus for replication [84]. Other receptor molecules, such as α5β1 integrin [85] and Ku80 autoantigen [86] have been proposed as potential coreceptors for B19V infection. However, the restricted uptake of B19V does not correspond with their wide expression profiles.

In an earlier study, we showed that the VP1u harbors a receptor-binding domain (RBD), which enables the uptake of the virus. Purified recombinant VP1u (recVP1u) was able to bind and to internalize exclusively into B19V permissive cells. Moreover, incorporation of VP1u subunits on bacteriophage VLPs by chemical coupling enabled their internalization into B19V permissive cells (Figure 2) [59]. The VP1u cognate receptor has not yet been identified, but its expression profile corresponds with the restricted tropism of B19V, being expressed exclusively in cells at erythropoietin-dependent erythroid differentiation stages [54,59].

### 5.2. Mapping and Structural Characterization of the Receptor-Binding Domain in the VP1u

The receptor-binding domain (RBD) in the VP1u was identified by using recVP1u variants with increasing N- and C-terminal truncations. The VP1u variants internalized normally when they were truncated less than 5 AA at the N-terminus or less than 147 AA at the C-terminus. Longer truncations at both ends decreased or blocked VP1u uptake [28]. According to these results, the RBD spans the region between AA 5–80 of VP1u, which explains the detectable exposure of this domain on the surface of susceptible cells before uptake [32,87], as well as the presence of a cluster of neutralizing epitopes [67,68].

The secondary structure analysis of the N-terminal of VP1u (AA 1–80) from natural B19V isolates, predicted a cluster of three α-helices with high confidence: helix 1 (AA 14–31), helix 2 (35–45), and helix 3 (59–68) (Figure 3A). However, only helix 1 was conserved among other erythroparvoviruses (Figure 3B) and displayed a prominent amphiphilic character. The marked segregation of polar and hydrophobic amino acids between the two opposite flanks of the α-helix is well suited for receptor binding. Compared with the residues of the hydrophilic side, the amino acids of the hydrophobic side were highly conserved (Figure 3C). Point mutations on the hydrophobic side blocked VP1u binding and internalization, suggesting a critical role of these residues in the interaction of VP1u with its cognate cellular receptor [28].

The sequence analysis of the first 80 amino acids of VP1u predicted two additional helices (Figure 3A). Disruption of the tertiary conformation of these domains by the introduction of flexible sequences strongly impaired VP1u internalization. This observation suggests that the spatial configuration of the three helices is crucial for VP1u binding to its cognate receptor and subsequent uptake. An ab initio modeling of the RBD by the QUARK algorithm [88] predicted a helix-like spatial configuration of the three helices (Figure 4A,B), where a cluster of conserved and internalization-relevant amino acids was modeled in close proximity (Figure 4C,D) [28]. The spatial proximity of function-relevant residues may correspond to a critical receptor-interacting site.

### 5.3. VP1u Cognate Receptor Facilitates B19V Targeting and Uptake Exclusively into Permissive Cells

B19V requires a strict intracellular environment for replication that can only be found in the erythroid progenitor cells (EPCs) in the bone marrow. The essential intracellular factors appear to be simultaneously upregulated in EPCs during the EPO-dependent differentiation stages. A study has shown that the internalization and the replication of B19V are considerably enhanced when CD34+ hematopoietic stem cells were stimulated with EPO [57]. Not surprisingly, the two cell lines that are most frequently used to study B19V infection, the megakaryocyte-erythroid UT7/Epo cells [89] and the erythroleukemic KU812Ep6 cells [90], are both derived from an EPO-dependent subclone. EPO signaling maintains the survival of cells that entered the intermediate erythroid differentiation stages [91,92,93]. Besides EPO signaling, B19V infection requires hypoxic conditions, which characterize the bone marrow microenvironment where the virus replicates. Hypoxia upregulates the signal transducer and activator of transcription 5 (STAT5) pathway, which facilitates viral DNA replication [61,94,95]. During the EPO-dependent differentiation stages, a cluster of erythroid-specific genes is upregulated [96], including the VP1u cognate receptor [59], which jointly are essential for B19V replication. In this regard, the main role of the VP1u receptor would be to facilitate the targeting and the uptake of B19V exclusively into cells providing a permissive intracellular environment for the infection. This strategy prevents the internalization into non-permissive cells, which would result in abortive infection.

### 5.4. Evolutionary Aspects of B19V Restricted Tropism and the Origin of the RBD in the VP1u

The origin of the marked tropism of B19V for erythroid precursors in the bone marrow is not known. The erythroparvovirus and dependoparvovirus genomes show striking similarities, both having identical hairpin telomeres at both sides, and related replication mechanism [97]. It is conceivable that the erythroparvovirus ancestor was dependent on helper virus co-infections. In line with this hypothesis, several studies observed the enhancement of B19V replication and gene expression in non-permissive cells in the presence of helper virus genes [98,99]. Besides the enhanced genome replication, adenovirus genes transactivated the B19V promoters, including the p44 promoter in the middle of the genome (nt 2247), which is normally silenced during B19V infection [100,101]. The p44 promoter is homologous to the promoters that initiate the expression of the structural capsid proteins in other parvoviruses. Interestingly, the expression of the structural proteins represents a limiting factor in B19V infection in non-erythroid cells. The transcription of the structural genes from the p44 promoter might have played an important role in the helper-dependent ancestors of erythroparvoviruses but could have been replaced during evolution by an alternative helper-independent replication in erythroid cells. In contrast to most other parvoviruses, B19V shows alternative splicing in the transcript from the p6 promoter that also enables the expression of the distal genes [102]. This exceptional splicing mechanism of B19V, which strikingly occurs only in EPCs, makes the internal and helper-dependent p44 promoter dispensable. Interestingly, there is another putative internal promoter (p55) at nt 2308 that might have similar properties.

According to this evolutionary model (Figure 5), the erythroparvovirus ancestor would have generally exhibited a helper virus-dependent replication in different tissues, and sporadically, a helper-independent replication in EPCs. However, without a specific targeting and internalization into the erythroid progenitor cells, the overall infection still depended on the helper virus co-infection. The erythroid-specific transcription of the structural genes from the p6 promoter generates a transcript with a longer 5′-UTR that possibly allows the displacement of the start codon of the capsid proteins and consequently, a longer VP1u region (Figure 6). The additional N-terminal amino acid sequence, expressed only during the helper-independent infection in erythroid cells, might have evolved to the RBD in the VP1u. The erythroid-specific targeting boosted the infection in the EPCs and thus represented a positive feedback loop that promoted the autonomous replication in the erythroid tissue. Vice versa, the helper-independent replication was the driving force for the positive feedback mechanism. The positive feedback enhanced the reliance on additional erythroid-specific factors and thus finally led to the extreme tropism of erythroparvoviruses.

The helper-dependent replication and expression of B19V genes in non-erythroid cells might still represent a significant aspect of the pathogenesis of B19V infection. The unspecific entry of the virus into non-erythroid tissues would not necessarily end in an abortive infection. The internalization of B19V during the late viremic phase by the antibody-dependent enhancement (ADE) provides a basis for latent infections in diverse tissues. These latent viruses might be sporadically reactivated by helper virus infections, which would explain many of the B19V-associated diseases as well as the recurrent detection of B19V DNA in the serum and different tissues [99,103].

## 6. Role of VP1u in the Subcellular Trafficking of Incoming B19V

To infect the cell, parvoviruses follow a complex route from the plasma membrane to the nucleus where they replicate. Various domains in the VP1u of parvoviruses have been shown to play a critical role in the process by assisting the transport of the incoming capsids throughout the different membrane-enclosed organelles and the highly crowded cytosol and by promoting their translocation through the nuclear pore complex (NPC) into the nucleus (Figure 6).

### 6.1. The Phospholipase A_2_ (PLA_2_) Domain

Following the interaction of the VP1u RBD with its cognate receptor, B19V is internalized by clathrin-mediated endocytosis and enters the endosomal pathway [104]. Endosomes provide cues that trigger capsid conformational rearrangements required for subsequent trafficking steps and contribute to the transport of incoming viruses to the nuclear vicinity. However, the mechanism followed by parvoviruses to escape from endosomal vesicles into the cytosol remains unclear. Phospholipase A_2_ enzymes (PLA_2_s) catalyze the hydrolysis of phospholipids and the release of lipid mediator precursors. Accordingly, PLA_2_s are key enzymes in many cellular processes such as lipid membrane metabolism, inflammation, membrane remodeling, host defense, and signal transduction [105]. PLA_2_s are found in mammalian tissues as well as in arachnids, insects, mollusks, reptiles, plants, and bacteria. A PLA_2_ domain containing the typical catalytic motif HDXXY and the calcium-binding site GXG is conserved in the VP1u of parvoviruses, including B19V (except for amdoparvoviruses) [21,22,23,25,106]. Mutations in either of these motifs disturbed both, the enzymatic activity and viral infectivity [20,21,23]. The pharmacological disruption of endosomal membranes or co-infection with endosomolytically active adenovirus, but not with inactive variants, partially rescued the infectivity of the PLA_2_ mutants, suggesting a role of the VP1u PLA_2_ in altering the endosomal membrane integrity to enable endosomal escape of viruses into the cytosol [24,26]. In B19V, VP1u mutations not related to the critical PLA_2_ motifs were also shown to reduce the enzymatic activity, probably by disrupting the three-dimensional rearrangement surrounding the PLA_2_ domain [107]. Although the PLA_2_ may facilitate the endosomal escape of incoming B19V, the mechanism involved is poorly understood. Interestingly, B19V endosomal escape was shown to occur without detectable endosomal membrane permeabilization or damage [104] and the enzymatic requirements for the PLA_2_ activity, i.e., pH and calcium concentration are not optimal in the endocytic compartment [106]. Accordingly, it remains unclear how the PLA_2_ activity of VP1u supports the escape of the endosomal capsids.

B19V PLA_2_ has been shown to up-regulate Ca^2+^ entry [108], to inhibit Na^+^/K^+^ ATPase activity and K^+^ channels [109,110], and to up-regulate ENaC [111]. These activities may contribute to the pathophysiology of B19V infections. Moreover, due to the inflammatory-like effects exerted by recombinant VP1u in cultured fibroblast [112] and in UT7/Epo cells [107], it has been hypothesized that the PLA_2_ domain of VP1u may contribute to B19V-associated syndromes, such as arthropathy and autoimmunity.

### 6.2. Nuclear Localization Signals (NLSs)

Following endosomal escape, parvovirus capsids are imported into the nucleus. Nuclear import of most proteins involves classical nuclear localization signals (NLSs) consisting of a stretch of basic amino acids, which interact with importin-α/importin-β to mediate transport through the nuclear pore [113]. The size of the parvovirus capsid is below the diameter limit of the nuclear pore complex (NPC). Therefore, capsids can theoretically be translocated through the NPC intact or without major disassembly [114]. NLSs have been identified in the VP1u from several parvoviruses [12,13,14,15,16,17,18,19] and confer nuclear import potential to the incoming particles via interaction with importin-β [115]. The VP1u of B19V does not display a motif resembling a classical NLS, however, when expressed in eukaryotic cells, VP1u accumulates in the cytoplasm and in the nucleus [107]. The stretch of basic amino acids found in VP2 occupies an internal position in the capsid and has been implicated in the nuclear translocation of assembly intermediates [116]. Accordingly, the mechanism of nuclear import of B19V remains uncertain and might differ fundamentally from that of other parvoviruses.

## 7. Biotechnological Applications of the VP1u of B19V

Nanocarriers are designed to efficiently deliver therapeutic molecules to specific tissues minimizing adverse effects [117]. Despite important progress, the drug delivery technology based on synthetic nanocarriers remains highly inefficient. One meta-analysis revealed that over 99% of the drugs do not reach the diseased cells and accumulate instead in non-target tissues or are cleared from the body [118]. Ideally, nanocarriers must specifically internalize into the target cells, escape from the endocytic compartment, and release their payload into the cytosol. These processes resemble the early infection steps of viruses, which operate as powerful natural nanocarriers to efficiently deliver genetic material into target cells by complex mechanisms shaped by evolution. The targeting machinery that is engaged in the early viral infection steps can be utilized to generate virus-inspired nanocarriers as efficient drug or gene delivery vehicles [119,120]. In this regard, the VP1u of B19V includes many interesting features that can potentially be exploited for drug delivery and diagnostics, i.e., specific cell targeting, efficient cell entry, and endosomal escape.

### 7.1. Specific Biomarker for EPO-Dependent Erythroid Differentiation Stages

Diverse hematological conditions (e.g., leukemia, thalassemic and myelodysplastic syndromes, bone marrow metastases of solid tumors, septicemia, or severe health conditions after surgery) are typically associated with the presence of erythroblasts outside the bone marrow [121,122,123,124,125,126]. Accordingly, the screening of peripheral blood for nucleated red blood cells (NRBCs) is used to recognize hematological disorders or severe health conditions. Assays to detect NRBCs must be very sensitive because the presence of only a few NRBCs can indicate serious underlying disorders. Unfortunately, automated hematology analyzers may not detect low levels of NRBCs. Besides, they generate suspect flags, which should be examined manually [127]. The currently used automated detection of NRBCs in peripheral blood has a detection limit of 1-2 erythroblasts per 100 white blood cells [123,128]. In comparison, VP1u decorated MS2 capsids were able to detect as few as one erythroleukemic UT7/Epo cell in 100,000 isolated white blood cells (unpublished observations). The sensitive identification of erythroblasts in the peripheral blood by fluorescent VP1u bioconjugates has the potential to improve the detection of diverse hematological disorders or severe health conditions and to facilitate an early diagnosis without the systematic need of an invasive technique such as bone marrow biopsy.

The precise identification and isolation of erythroid progenitor cells is important in hematological research and in diagnostics to characterize and treat bone marrow disorders. However, the technique remains rather complex and laborious, since the currently used markers are not lineage-specific (CD36, CD38, CD44, CD45, CD71, CD105, EPOR) or are broadly expressed during the erythroid development (glycophorin A). Therefore, the combination of several antibodies is necessary to achieve the correct identification [124,129,130,131,132,133,134]. In contrast, the fluorescent VP1u bioconjugate appeared as a unique and highly sensitive marker for the EPO-dependent erythroid differentiation stages and readily detected these cells in heterogeneous cell populations from different tissues [54]. The findings show the potential of the VP1u as a biomarker to identify and sort erythroid differentiation stages in a simpler procedure than it has been practiced so far.

It is expected that the future biotechnological applications of the VP1u will be spurred by the identification of its cognate receptor. However, the identity of the VP1u receptor will not necessarily be determinant for the applicability of the VP1u as a specific cellular marker. Historically, it is not uncommon to use cell surface markers to identify cell populations based on empirical evidence without knowing the identity and/or the function of the targeted receptors.

### 7.2. Specific Drug Delivery and Chemotherapy

#### 7.2.1. β-Hemoglobin Disorders

β-hemoglobin disorders are a group of highly prevalent hereditary diseases caused by mutations in the gene encoding for the β-chain of hemoglobin, resulting in qualitative and quantitative defects in β-globin production. β-thalassemias are a heterogeneous group of genetic disorders characterized by the partial or complete absence of β-globin chain production, leading to anemia and iron overload. The disease is highly prevalent with 80–90 million carriers worldwide. Without diagnosis and appropriate treatment, the severe forms of β-thalassemia lead to death before age 20 [135]. Sickle cell disease (SCD) is the most common and severe hemoglobinopathy. In SCD, a single mutation in the β-globin gene results in the production of an aberrant hemoglobin molecule, which causes the rigid sickle-like shape of erythrocytes. Without treatment, SCD is lethal before age five [136].

Patients with severe β-hemoglobin disorders require regular blood transfusions, which lead to iron overload and related complications. Accordingly, iron chelation therapies are also required [137,138]. The most severe forms of the disease have been successfully treated by allogeneic hematopoietic stem cell transplantation from a matched related donor. However, major drawbacks are the difficulty to find a histocompatible donor and the need for extensive immunosuppressive regimens, with the risk of immunological complication. Besides, this approach is not accessible for many affected individuals [139,140]. Gene therapy and gene editing strategies to restore the globin genes have generated promising results. However, these approaches lack cell-specific vectors, resulting in poor efficiency and the risk of insertional oncogenesis [141,142,143].

Due to the numerous drawbacks associated with the current therapeutic strategies, there is a great interest in developing novel therapeutic options. The therapeutic targeting of RNA by double-stranded RNA-mediated interference (RNAi) or by antisense oligonucleotides (ASOs) allows specific inhibition of the target of interest and a very rapid transferability to the clinics [144]. However, the delivery of nucleic acid molecules to the bone marrow remains highly inefficient. The MS2 capsid is a well-studied vector for drug delivery and can be easily loaded with therapeutic ASOs or small interfering RNAs (siRNA) [145,146]. This strategy provides protection of the therapeutic nucleic acid molecules in the extracellular milieu, avoids solubility problems, and thus allows more the options to improve the modifications of the oligonucleotides. In a previous study, we showed that anchoring of VP1u subunits to the surface of MS2 capsids retargets the particles to erythroid cells. This finding offers the opportunity to deliver encapsidated genetic material specifically to this cell population [59]. Potential targets of therapeutic ASOs or siRNA might be different factors involved in the regulation of erythropoiesis, such as transferrin receptor 2, or regulatory elements of fetal hemoglobin, such as B-cell lymphoma/leukemia 11A and erythroid Kruppel-like factor. Specific downregulation of such factors in erythroid progenitor cells would significantly alleviate symptoms of β-hemoglobin disorders [147,148,149,150].

#### 7.2.2. Erythroleukemia

Acute erythroleukemia is a rare disorder associated with a poor prognosis. A study reported a median overall survival of 8 months [151]. The treatment of erythroleukemia is compromised due to the systemic distribution and resistance of the malignant cells to chemotherapeutics [152,153]. Therefore, the successful elimination of erythroleukemic cells by a cytotoxin requires a “magic bullet” strategy—an efficient and specific targeting of the toxin to cancer cells—minimizing adverse effects to the surrounding healthy cells [154]. Erythroleukemias exhibit proliferating cancer cells in the early and intermediate erythroid differentiation stages [155], which are the target cells of the VP1u. Accordingly, the VP1u-mediated toxin delivery represents a possible strategy to overcome the resistance of erythroleukemia to chemotherapeutics. In previous studies, VP1u successfully targeted a toxin specifically to malignant erythroid precursors and thus selectively eliminated these cells from a mixed cell culture [156].

The immunity of many individuals against B19V would represent a serious obstacle for the application of the VP1u-targeted delivery. About half of the human population is seropositive for anti-B19V antibodies. Similar problems are faced in the application of AAV vectors for gene therapy, where many individuals have antibodies against serotypes 2 and 3 [157]. Following the natural mechanism of viruses to evade the immune system, the AAV researchers are searching for AAV isolates and isotypes, which are not neutralized by the common pool of antibodies, but still offer the beneficial properties of the original virus [158,159,160]. In the case of a short protein with a single function as with the RBD of the VP1u, an immune escape by antigenic drift is easier to achieve without disturbing the receptor binding and internalization capacity. The natural mutations observed in various B19V isolates (Figure 3C) together with the mutational studies already performed [28], provide an excellent basis to mimic an antigenic drift of the VP1u RBD without decreasing the targeting function of the protein. Furthermore, there exist different options to reduce the antigenicity of a therapeutic protein, such as a fusion with an abundant endogenous protein as serum albumin or the immunoglobulin constant fragments [161,162,163]. The coupling to these endogenous proteins does not only circumvent the immune response, but also considerably increases the solubility, stability, and serum half-life of the therapeutic proteins. In line with this concept, bovine serum albumin (BSA) was used as an adaptor molecule for the attachment of the toxins to a VP1u-NeutrAvidin complex. The results showed that the modified BSA remained soluble after the attachment of 20–30 fluorescein or toxin molecules to the protein and was targeted exclusively to VP1u-expressing cells. The stability of the drug attachment might be increased by packing the effector molecule into a capsid, as shown with the MS2 bacteriophage in previous studies [145,146,164]. The specific delivery of an encapsidated effector allows a higher dose per delivered particle without increasing toxicity. Besides, the capsid can be engineered to incorporate multiple residues to improve the targeting efficiency.

## 8. Concluding Remarks

The VP1u is a key component of the capsid of human parvovirus B19 with essential functions in multiple steps of the infection, such as tissue tropism, uptake, intracellular trafficking, and entry. The VP1u is also the immunodominant region of the capsid and a crucial component for prospective vaccines. In the future, efforts will be focused to better understand the essential functions of VP1u in B19V infection and to identify the VP1u interactome, notably its cognate cell receptor.

Recent innovations in protein engineering and nanomaterials science have the potential to revolutionize the conventional methods of diagnosis and treatment, bringing new hopes to patients. However, to date, a major barrier in their clinical application remains their poor selective targeting. Only a few clinically approved nanoscale delivery vehicles integrate molecules to selectively target the cargo to the tissue of interest. In this regard, the remarkable erythroid specificity of the VP1u offers novel opportunities to generate virus-inspired biomarkers and nanocarriers to specifically target erythroid cells. This approach may contribute to a better understanding of the mechanisms governing erythroid development and to treat disorders of the erythroid lineage. Efforts to circumvent the VP1u immune response and to optimize the stability and density of cargo delivery will facilitate its transferability to human diagnostics and therapies.

## Figures and Tables

**Figure 1 viruses-12-01463-f001:**
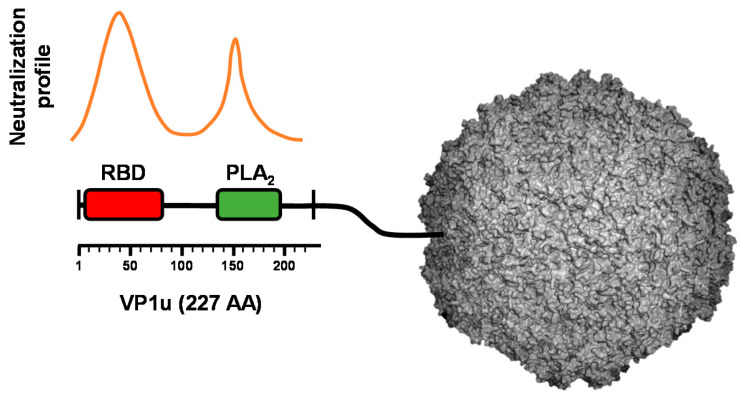
Schematic depiction of the neutralization profile and functional domains in the VP1u of B19V. The neutralizing profile revealed a cluster of important epitopes in the N-terminal region of the VP1 corresponding to functional domains. RBD; receptor-binding domain. PLA_2_; phospholipase A_2_ domain.

**Figure 2 viruses-12-01463-f002:**
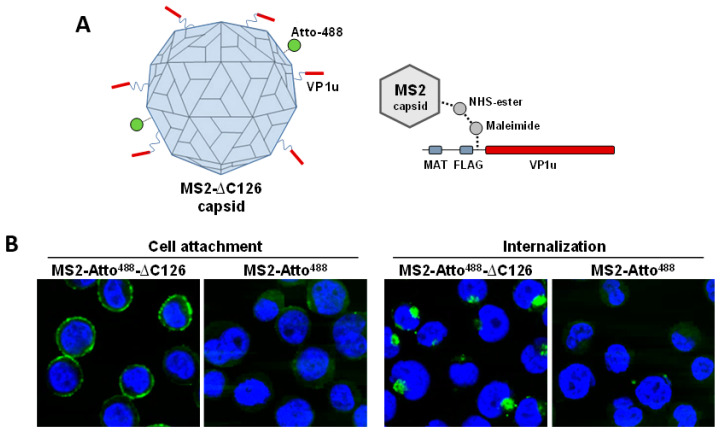
Binding and internalization of recombinant MS2-VP1u labeled with Atto-488. (**A**) Schematic depiction of MS2-VP1u particles. (**B**) Confocal fluorescence microscopy of MS2-VP1u bound to UT7/Epo cells at 4 °C and internalized at 37 °C. MS2-Atto^488^-Δ126 (100 N-terminal AA of VP1u); MS2-Atto^488^ (without VP1u).

**Figure 3 viruses-12-01463-f003:**
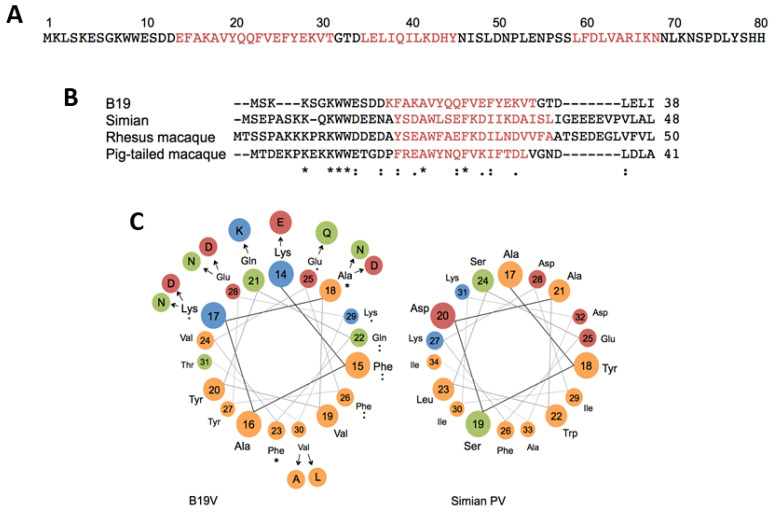
Structural motifs within the VP1u RBD. (**A**) Amino acid sequence of the N-terminal of VP1u (AA 1–80) and the three predicted alpha helices (underlined red). (**B**) Conservation of alpha helix 1 among erythroparvoviruses. (**C**) Modeled helical wheel of the conserved helix 1 (AA 14–31) shows the spatial arrangement of hydrophobic and polar amino acids within helix 1. Amino acid differences found in B19V isolates are shown in a wider radius. Hydrophobic = orange; polar = green; basic = blue; acid = red. The helical wheel of the simian parvovirus helix 1 is shown.

**Figure 4 viruses-12-01463-f004:**
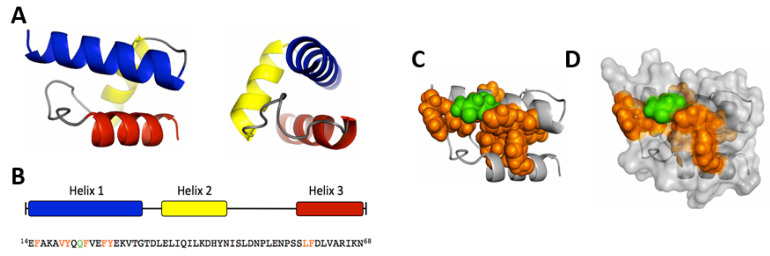
Ab initio modeling of the RBD in the VP1u. (**A**) Front and side views. Helix 1 appears in blue (AA 14–31), helix 2 in yellow (AA 35–45), helix 3 in red (AA 57–68). (**B**) Helix distribution and sequence of the modeled AA 14–68. The amino acids required for VP1u internalization are colored in orange (hydrophobic) and green (polar). (**C**) The spatial distribution of essential amino acids is shown as spheres in the helical structure (**D**) and in the surface model of the RBD.

**Figure 5 viruses-12-01463-f005:**
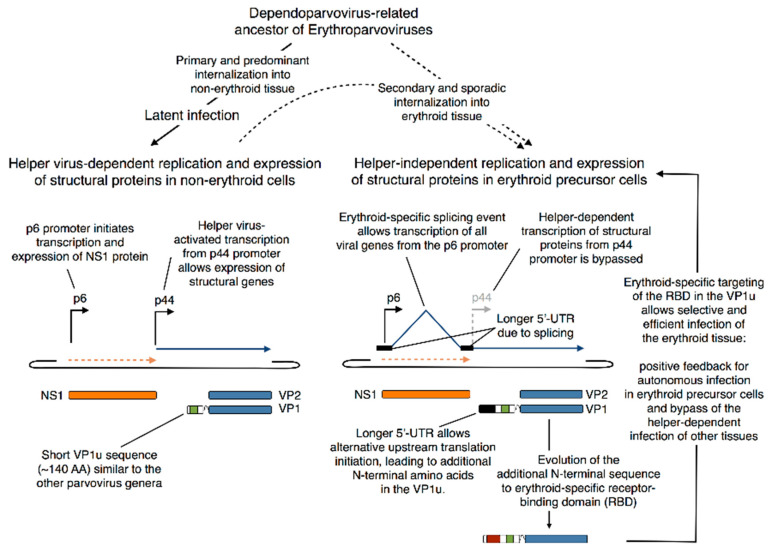
Proposed origin of the marked erythroid tropism of B19V (see text for details).

**Figure 6 viruses-12-01463-f006:**
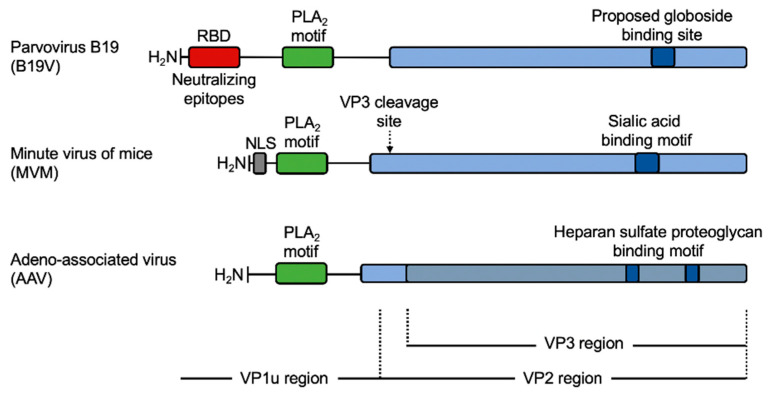
Infection-relevant functional domains in the structural proteins of three representative parvoviruses. B19V exhibits a longer VP1u region compared to other parvoviruses. The additional N-terminal stretch of 80–90 amino acids contain the functional RBD [28] and a cluster of neutralizing epitopes [67,68]. NLS, nuclear localization signal.

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
