# Peer review of "The VP1u of Human Parvovirus B19: A Multifunctional Capsid Protein with Biotechnological Applications"

_viruses, 2020, doi:10.3390/v12121463_

Round 1
Reviewer 1 Report
The article is well written and the VP1u of parvovirus B 19 is described in great detail.
Here are several remarks:
Line 103: Four species of human bocaviruses are currently known and included in the Bocavirus genus. Please clarify it.
Figure 2. Could you add the proper bar scale?
Missing citing references for Figure 6.
Lines 130-132: Do the recombinant parvoviral particles remain VP1u?
Author Response
Reply to reviewer 1
Line 103: Four species of human bocaviruses are currently known and included in the Bocavirus genus. Please clarify it.
(HBoV) has been changed to (HBoV1-4).
Figure 2. Could you add the proper bar scale?
These are immunofluorescence images of methanol/acetone fixed cells. In our opinion, the size of the cells is irrelevant.
Missing citing references for Figure 6.
References have been included.
Lines 130-132: Do the recombinant parvoviral particles remain VP1u?
It is written "VP2 recombinant particles". Accordingly, these particles lack the entire VP1 (including VP1u).
Reviewer 2 Report
In this extensive review the authors give a general overview on Parvoviridae family, structure and replication, paying particular attention to human parvovirus B19 (B19V). Role of VP1u in restricted tropism of B19V as well in the subcellular trafficking of incoming virus are described in details, so giving explanations and background for the last section – biotechnological applications of the VP1u of B19V in diagnostic, drug delivery and chemotherapy of hematological diseases. The article is interesting and despite the fact that the material is relatively complex (especially for those who are not specialists in the field of parvoviruses) it is written in an understandable way and explained by data from previous publications.
In reality I have only one minor comment:
Line 248 – “Figure 4C, D and E” – there is no 4E in the Figure 4 (Lines 250-254).
Author Response
Reply to reviewer 2
Line 248 – “Figure 4C, D and E” – there is no 4E in the Figure 4 (Lines 250-254)
Error has been corrected